# Surgery for Adult Patients with Congenital Heart Disease: Results from the European Database

**DOI:** 10.3390/jcm9082493

**Published:** 2020-08-03

**Authors:** Vladimiro L. Vida, Lorenza Zanotto, Laura Torlai Triglia, Lucia Zanotto, Bohdan Maruszewski, Zdzislaw Tobota, Francesco Bertelli, Claudia Cattapan, Tjark Ebels, Daniele Bottigliengo, Dario Gregori, George Sarris, Jurgen Horer, Giovanni Stellin, Massimo A. Padalino, Giovanni Di Salvo

**Affiliations:** 1Paediatric and Congenital Cardiac Surgery Unit, Department of Cardiac, Thoracic and Vascular Sciences, University of Padua, 35128 Padua, Italy; lorenza.zanotto@gmail.com (L.Z.); laura.torlait@gmail.com (L.T.T.); francesco.bertelli.2@studenti.unipd.it (F.B.); claudia.cattapan@gmail.com (C.C.); giovanni.stellin@unipd.it (G.S.); massimo.padalino@unipd.it (M.A.P.); 2Department of Statistical Sciences, University of Padua, 35121 Padua, Italy; lucia.zanotto@unive.it; 3Department for Pediatric Cardiothoracic Surgery, The Children’s Memorial Health Institute, 04730 Warsaw, Poland; bmar@pol.pl (B.M.); ztobota@ecdb.pl.pl (Z.T.); 4Department of Cardio-Thoracic Surgery, University Medical Center Groningen, 9713GZ Groningen, The Netherlands; tjarkebels@mac.com; 5Unit of Biostatistics, Epidemiology and Public Health Unit, Department of Thoracic, Cardiac and Vascular Sciences, University of Padua, 35131 Padua, Italy; daniele.bottigliengo@unipd.it (D.B.); dario.gregori@unipd.it (D.G.); 6Athens Heart Surgery Institute and Department of Pediatric and Congenital Cardiac Surgery, Iaso Children’s Hospital, 15125 Athens, Greece; gsarris@mac.com; 7Department for Congenital and Pediatric Heart Surgery, German Heart Centre Munich, 80636 Munich, Germany; hoerer@dhm.mhn.de; 8Pediatric Cardiology Unit, Department of Children and Woman’s Health, University of Padua, 35128 Padua, Italy; giovanni.disalvo@unipd.it

**Keywords:** congenital heart disease, adults, surgery, hospital outcome

## Abstract

Adults with congenital heart disease (ACHD) represent a growing population. To evaluate frequency, type and outcomes of cardiac surgery in ACHD, we gathered data from the European Congenital Heart Surgeons Association Database of 20,602 adult patients (≥18 years) with a diagnosis of congenital heart disease who underwent cardiac surgery, between January 1997 and December 2017. We demonstrated that overall surgical workload (as absolute frequencies of surgical procedures per year) for this specific subset of patients increased steadily during the study period. The most common procedural groups included septal defects repair (*n* = 5740, 28%), right-heart lesions repair (*n* = 5542, 27%) and left-heart lesions repair (*n* = 4566, 22%); almost one-third of the procedures were re-operations (*n* = 5509, 27%). When considering the year-by-year relative frequencies of the main procedural groups, we observed a variation of the surgical scenario during the last two decades, characterized by a significant increase over time for right and left-heart lesions repair (*p* < 0.0001, both); while a significant decrease was seen for septal defects repair (*p* < 0.0001) and transplant (*p* = 0.03). Overall hospital mortality was 3% (*n* = 622/20,602 patients) and was stable over time. An inverse relationship between mortality and the number of patients operated in each center (*p* < 0.0001) was observed.

## 1. Introduction

Advances in pediatric cardiac care have led to surging survival rates for patients with congenital heart disease (CHD). Consequently, the number of adults (≥18 years of age) with congenital heart disease (ACHD) has significantly grown over time, reaching in 2010 two thirds of the CHD population [1]. Many patients require repeated cardiac surgery after primary repair in infancy for residual lesions or complications, while other patients with simple CHD, not diagnosed during early infancy, some in the setting of a complex secondary physiology, may require surgical treatment later in life [2]. The aim of this study was to assess the frequency, type and outcomes of cardiac surgery in adults with CHD during the last 20 years, within the European Congenital Heart Surgeons Association (ECHSA). The year-by-year relative frequencies of the most common procedural groups were also evaluated.

## 2. Materials and Methods

### 2.1. Data Source

This is a retrospective longitudinal study. Data were gathered from the European Congenital Heart Surgeons Association Database (ECHSA-DB). The study was performed under the aegis of the ECHSA and was approved by the ECHSA Scientific Committee. The procedures followed were in accordance with record review and protection of patient confidentiality.

As of December 2017, the ECHSA-DB contained de-identified data on 267,567 operations, performed in 393 centers from 83 countries worldwide. A total of 21,991 procedures performed on adult patients with CHD were recorded. Preoperative, operative data and outcomes were collected on all patients undergoing congenital heart operations in all participating centers. Data were inserted into the ECHSA-DB by registered cardiothoracic surgery centers and submitted using the International pediatric and congenital cardiac code in the version of the International Congenital Heart Surgery Nomenclature and Database Project of the European Association for Cardio-Thoracic Surgery and the Society of Thoracic Surgeons. The ECHSA-DB characteristics and data verifications modalities have already been reported [3,4,5].

### 2.2. Patient Population

In this study, we enrolled all patients beyond the age of 18 years with a diagnosis of CHD, who underwent cardiac surgery between January 1997 and December 2017 and had a complete minimal dataset (Appendix A). Patients with isolated acquired cardiac disease or receiving interventional cardiology procedures were excluded (*n* = 1869 patients).

We identified a total of 279 different surgical procedures. According to the ECHSA-DB classification, we categorized each procedure within procedural subgroups. Outcome analysis was performed according to the main surgical procedure (procedure leading to surgery) [2]. In the ECHSA-DB each patient is associated with a unique deidentified ID code; this allowed us to evaluate if the same patient had multiple cardia procedures during the considered study period. In the presence of multiple concomitant procedures, the procedure with the highest mortality risk score was considered as the leading surgical procedure, according to Fuller’s proposed ACHD score [6].

In the case of extracorporeal membrane oxygenation (ECMO) or ventricular assist device (VAD) implantation only, it was considered as leading surgical procedure when isolated, while it was considered as associated procedure when performed with other surgical procedures.

The primary aim of the study was to evaluate frequency, type and outcomes, in terms of hospital mortality, of cardiac surgery in ACHD. Hospital mortality was defined as any death occurring during hospitalization or within 30 days after surgery. The secondary aim was to assess any variations in the number of procedures for each of the procedural groups (as relative frequencies computed year-by-year), during the study period.

### 2.3. Statistical Analysis

Categorical variables were presented as percentage, and continuous variables were expressed as median, with interquartile range (IQR) as a measure of variability. To investigate correlations among variables, univariate analysis was performed with Fisher’s exact test and Wilcoxon rank test for nominal and continuous variables, respectively. Given the multicenter nature of the study, in order to clarify the relationship between center volume and mortality we performed a logistic regression using the proportion of deaths as outcome variable and the number of operated patients as independent variable. To take into account this type of relation in the multivariate analyses we decided to use multilevel logistic models: fixed effects were assigned to the statistically significant variables of interest, while random effects were estimated for the “center effect”. For each procedural group, line graphs display variations in frequency and mortality over the study period. The pattern can be seen as time series and consequently analyzed with an autoregression model, which represented the most appropriate technique for processing our data, after observing auto correlation function (ACF) and partial auto correlation function (PACF). The same methodology was employed to study the year-by-year evolution of the relative frequencies of specific procedures. In order to better visualize the path of the time series, we used a local polynomial regression to smooth its trend. The probability of the first type error was set at α = 0.05 except for the univariate analyses where the threshold was α = 0.25 in order to reduce selection bias [7]. We examined data with R version 3.4.0 statistical package [8].

## 3. Results

### 3.1. Population Characteristics

In our study, we included 20,602 patients from 152 ECHSA centers whose data were entered in the ECHSA-DB. Median age at surgery was 33 years (interquartile range 23–47 years) and 10,464 (51%) were males; cohort’s median age remains stable during the observational period considered (Figure 1). Septal defects repair was the most represented procedural group (28%), followed by right-heart lesions repair (27%) and left-heart lesions repair (22%). Almost one-third of the procedures consisted of re-operations (*n* = 5508, 27%); a total of 984 patients (4, 8%) underwent multiple surgical procedures during the study period. Patients’ characteristics, surgical variables and outcomes, according to surgical procedural groups, are listed in Table 1 and Table 2. A total of 7564 patients (37%) required 10,678 associated surgical maneuvers at the time of the main surgical procedures (Table 3).

### 3.2. Surgical Trends

Overall surgical workload (as absolute frequencies of surgical procedures per year) increased steadily during the study period (Figure 2) (Appendix A). However, when considering the year-by-year relative frequencies of the main procedural groups, we observed a significant increase of right and left-heart lesions repairs (*p* < 0.001 for both) (Figure 3); while a significant decrease was seen in regards to septal defects repair (*p* < 0.001) and transplants (*p* = 0.03) (Figure 4). As of thoracic arteries and veins anomalies repair, partial anomalous pulmonary venous connection (PAPVC) repair, single ventricle associated-procedures, mechanical support implantation and other less common surgical procedures, their relative frequency remained stable over the study period.

### 3.3. Hospital Mortality

Overall hospital mortality rate was 3% (*n* = 622/20,602 patients) and it remained stable over the study period (*p* = 0.2) (Appendix A). Some operations carried a different surgical risk whether performed isolated or in association to other procedures, the two different operative risks for each procedure are shown in Appendix A. Surgical risk was found to be particularly high for some procedural groups such as transplants (19%) and single ventricle associated-procedures (15%). A logistic regression using the proportion of deaths as outcome variable and the number of operated patients as independent variable was performed; the results showed the slope was negative and statistically significant. We demonstrated an inverse relationship between hospital mortality and the overall number of patients operated in each hospital, what we called “center effect” (hazard ratio = 0.9997, *p* < 0.0001) (Figure 5). These results show that the probability of fatal outcome decreases with increasing of operated patients and expertise (i.e., volume center). At multivariate analysis, hospital mortality was significantly related to a longer cardiopulmonary bypass (CPB) (hazard ratio = 1.01, *p* < 0.0001), lower patient’s body surface area (BSA) at the time of surgery (hazard ratio = 0.39, *p* < 0.001) and to the need for reoperation (hazard ratio = 1.94, *p* < 0.001). The association between the various risk factors and hospital mortality, in the multivariate analysis, was adjusted considering the “center effect”.

## 4. Discussion

The incidence of CHD is approximately eight per 1000 live-born children, about 85% of these patients are nowadays reported to survive into adulthood [9,10,11,12,13]. The increased survival of patients with CHD can be attributed to improvements in imaging techniques that increased the chances of detecting prenatal and postnatal cardiac anomalies, as well as the optimization of the results of early correction in neonates and infants and advances in pre- and postoperative care, in addition to the widespread screening of asymptomatic patients. All these factors also contributed to modify the spectrum of patients with CHD reaching adulthood also extending the survival after the second decade of life for most of the patients with more severe CHD (i.e., hypoplastic left heart syndrome, tetralogy of Fallot-pulmonary atresia, etc.).

Adult population with CHD generally includes: (A) patients with previously undetected CHD who remained asymptomatic during childhood (i.e., atrial septal defects—ASD), (B) patients with increasing cardiac symptoms not controlled by medical therapy (i.e., patients with recognized CHD who have survived into adult age without any surgical treatment and without developing irreversible damages to their heart and lungs; patients who are candidates for correction after previous palliative procedures; patients who either require reoperations because of late complications or residual defects and patients who require heart transplantation for late ventricular failure after repair or single ventricle palliation).

According to the results of this study, we can affirm that the surgical workload determined by the treatment of adult patients with CHD increased steadily over time. Similar to previously reported data from ECHSA and other studies [14,15,16,17,18], septal defects repair, right-heart lesions and left-heart lesions repair continue to be the three most frequent surgical categories of lesions in this subset of patients. When considering the relative surgical trends over the study period, we were able to demonstrate a changing surgical scenario over the last two decades, characterized mainly by a significant increase in the number of patients requiring right and left-heart lesions repair, while a statistically significant decrease was observed for septal defects repair and transplants.

A possible explanation of such a decreasing trend may derive from the recent important achievements in interventional cardiology, which could also help explain the year-by-year decrease in relative frequencies of specific surgical procedures, such as ASD closure and coarctation repair [19,20,21,22]. Another factor contributing to the decrease in septal defect closure is the higher number of defects diagnosed and treated earlier in childhood due to the advances in imaging techniques (i.e., echocardiography).

On the other hand, the decrease of transplants may be due to multiple factors. First, the dramatic change in donors’ and recipients’ characteristics, donors’ cause of death is nowadays most commonly ischemic brain injuries than trauma, therefore they are more likely to be older and to present with more comorbidities, which can affect the quality of the graft. The scarcity of suitable grafts and the high rate of family refusal to donate associated with the difficulties in optimal organ allocation have also influenced greatly the feasibility of heart transplant. Recently, given the limited availability of donors, VADs are currently implanted as destination therapy more often. In addition, patients who are unsuitable for a transplant or who would not survive the waiting time are now candidates for VADs, even though in our series they were not common [23].

In this study, reoperations accounted for almost one-third of the overall number of surgical procedures, which is comparable to other published reports [19,20,21,22]. It is interesting to note that with a growing ACHD population, reoperations were expected to outnumber primary repairs [9]; however, as stated above, we believe that the increasing number of interventional procedures performed by cardiologists (i.e., pulmonary valve implantation, coarctation repair and ASD closure) seems to keep their number fairly stable overtime [12,24].

Overall hospital mortality for ACHD requiring surgical treatment ranges from 1.3% to 6.8% [12,25]. In our analysis, the most striking result was that the risk for cardiac surgery in the entire population remained, indeed very low, 3% [6,12,25,26] and relatively stable during the last 20 years. Nonetheless, hospital mortality continues to be high for particular procedural subgroups such as transplants or single-ventricle associated procedures [6,21,27]. Based on our data we were able to demonstrate that hospital mortality was significantly related to re-operations, which can be considered both as a marker of a complex disease but also as an independent risk factor for surgical outcomes and the complexity of the surgical procedure (i.e., the association to other CHD requiring concomitant surgical treatment, which contributed to the prolongation of CPB and cross-clamp times). In our series also smaller BSA was associated with an increased mortality risk, especially in the context of left heart lesions repair, which comprises mainly valvular procedures. Smaller BSA is a well-known risk factor for worse surgical outcomes, when associated with smaller valvular annulus size and the need of smaller valvular prosthesis, it could account for the increased mortality risk.

In order to better characterize the surgical risk of the single procedures, we determined the difference in operative risk whether they were performed as isolated or in association. Of note, tricuspid valve repairs carried a higher surgical risk when performed as isolated procedures, as well as pulmonary valve repairs; on the other hand, right ventricle outflow tract (RVOT) procedures and others showed a higher operative risk when performed in association (Appendix A).

In addition, we demonstrated an inverse relationship between in-hospital mortality and the number of operations performed by single centers (“center effect”). Therefore, whenever feasible, must consider that ACHD patients need be referred to high-volume and highly specialized ACHD surgery centers, which carry a lower surgical risk [15,28,29].

Several scores were created for estimating the risk of surgery in children with CHD [23,28]. On the other hand, up-to-date there is no largely validated specific risk score for cardiac surgery in adult patients with CHD. We strongly believe that, in order to establish a comprehensive score for the evaluation of surgical risk in this growing population of adult patients, it is mandatory to take into account associated comorbidities, risk factors, the complex pathophysiology in which a procedure is performed and basic patient’s diagnosis together with the diagnosis leading to surgery.

### Limitations

This is a retrospective data examination over a long period of time and therefore an inter-institutional and intra-institutional variability of surgical treatment cannot be excluded. Not all ECHSA centers participated to data collection, in addition some adult patients with CHD could have been lost at follow-up by the original congenital cardiac center where they were first, diagnosed, or could have been operated in cardiac surgery programs; all these factors could have contributed to underestimate the real number of patients who required surgery for treating their CHD in the adult age.

ECHSA-DB is specifically based on surgical centers which deal patients with CHD, therefore the number of patients receiving interventional cardiology procedures which was excluded from this study is certainly underestimated.

Furthermore, for this manuscript we decided to use the minimal ECHSA database that was 100% completed, to give more consistent data on our goals, as to describe the frequency, types and outcomes (as rough mortality) of cardiac surgical procedures on adult patients with CHD. Other variables (as preoperative and postoperative) are present in the ECHSA database but can be uncompleted and were not considered since they could affect statistical consistency.

Another limitation may be represented by the fact that we only used a minimal data set. This could have excluded significant patient’s related variables, which could have influenced surgical outcome. In addition, regarding the determination of the “center effect” significant patient’s related variables which were not available, in particular the complexity of the procedure, could have influenced surgical outcome.

## 5. Conclusions

We report data from the largest series of adult patients with CHD who required surgical treatment hitherto evaluated. We were able to evaluate the trend of the surgical workload over the years for this specific subset of patients, indicating a progressive increase over time. We also demonstrated a variation of the surgical scenario during the last two decades, characterized by an increase in right and left-heart lesions repairs and a significant decrease in septal defects repair and transplants. We concluded that operative mortality in ACHD patients remained low and stable over the last two decades, but some procedures continued to carry high mortality rates.

Due to the inverse relationship between mortality and the number of patients operated in each hospital, these patients need to be centralized, whenever feasible, in specialized centers with experience in ACHD care.

Our data can serve as a benchmark to validate current available risk score for this growing population of adults requiring cardiac surgery.

Further analysis including all interventional procedures (hemodynamical and arrhythmia procedures) could substantially contribute to better determine the optimal treatments and outcomes for ACHD patients.

## Figures and Tables

**Figure 1 jcm-09-02493-f001:**
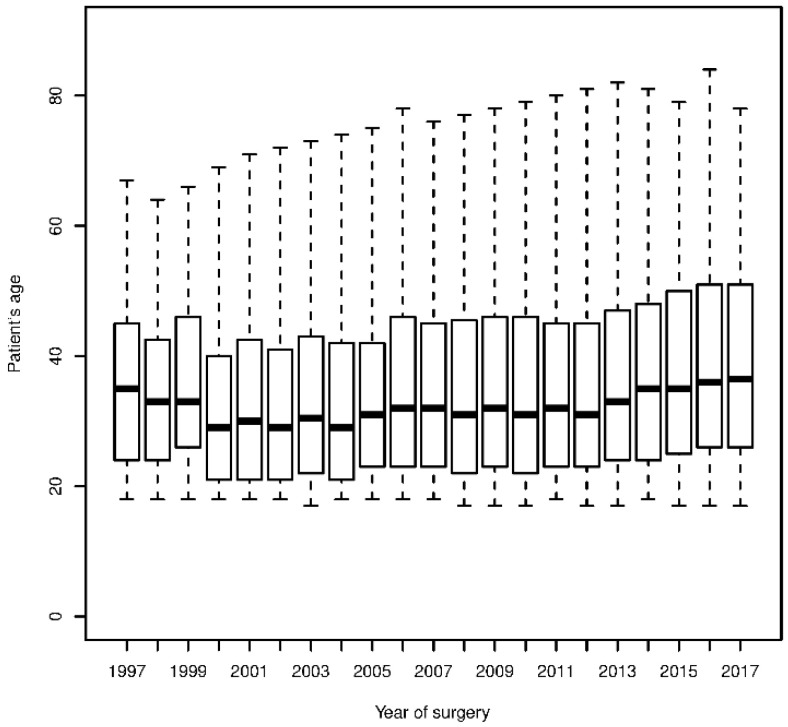
Boxplot showing variation of patients’ age at surgery during the last two decades. Results are shown as median age, interquartile range and overall interval.

**Figure 2 jcm-09-02493-f002:**
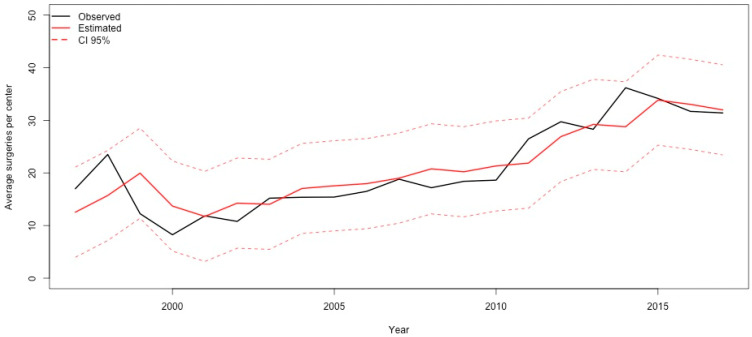
Time–series showing the observed (solid black line) and estimated (solid red line) average surgical workload according to the year of surgery in adult patients with congenital heart disease. Data shown as the average number of procedures per center. The 95% confidence interval is reported as dashed red line.

**Figure 3 jcm-09-02493-f003:**
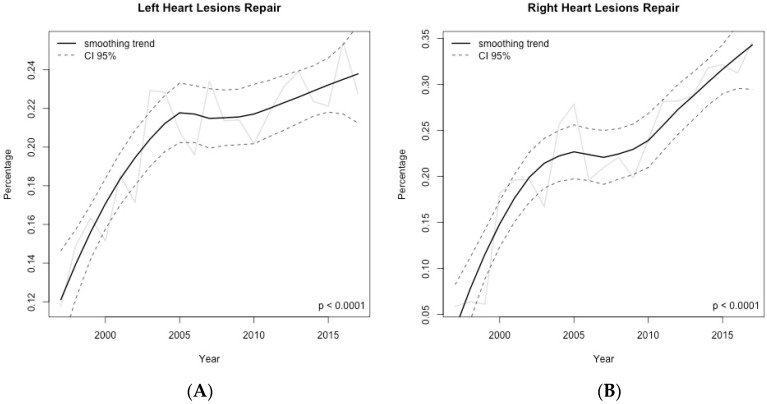
Time–series of the relative frequencies computed year-by-year for the main procedural surgical groups in adult patients with congenital heart disease. (**A**) left heart lesions repair; (**B**) right heart lesions repair. Data shown as rough number of the observed relative frequencies (solid black line) and smoothed trend (solid gray line) with 95% confidence interval (dashed black line).

**Figure 4 jcm-09-02493-f004:**
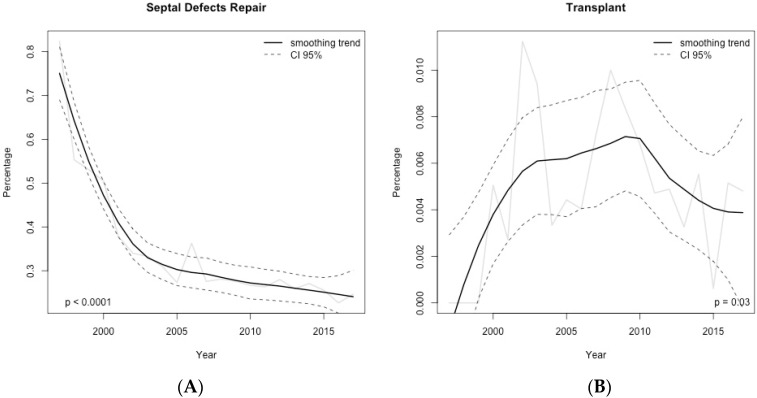
Time–series of the relative frequencies computed year-by-year for the main procedural surgical groups in adult patients with congenital heart disease. (**A**) septal defects repair; (**B**) transplants. Data shown as rough number of the relative frequencies (solid black line) and smoothed trend (solid gray line) with 95% confidence interval (dashed black line).

**Figure 5 jcm-09-02493-f005:**
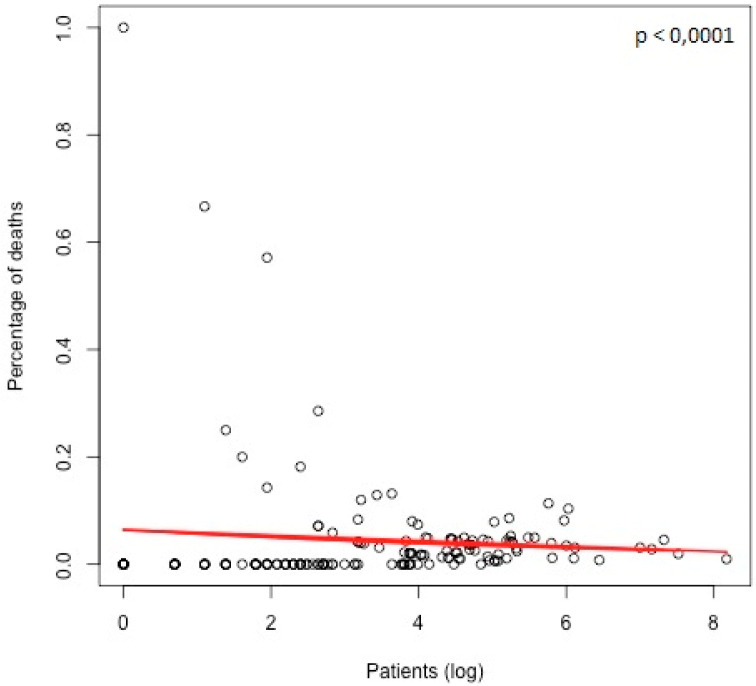
Graphic demonstration of the relationship between center volume (as patients, plotted in logistic scale) and the percentage of death. A significant inverse relationship was found between the two variables. The red line represents the logistic regression model estimated.

**Table 1 jcm-09-02493-t001:** Demographics and patient characteristics.

Variable	Total *	Males *	Age (years) **	BSA **	CC Time **	CPB Time **	Reoperation *	Hospital Mortality *
Total	20,602	10,464 (51)	33 (23–47)	1.8 (1.6–1.9)	64 (38–100)	92 (56–141)	5508 (27)	622 (3)
Septal defects repair	5740 (28)	2360 (41)	35 (25–48)	1.7 (1.6–1.9)	39 (25–63)	67 (47–100)	208 (3.6)	56 (1)
Right-heart lesions repair	5542 (27)	2751 (49)	31 (23–44)	1.7 (1.5–1.9)	62 (41–90)	99 (71–140)	2113 (38)	166 (3.0)
Left-heart lesions repair	4566 (22)	2810 (62)	34 (23–50)	1.8 (1.7–2.0)	91 (65–127)	128 (94–179)	1385 (30)	124 (2.7)
Thoracic arteries and veins anomalies repair	1522 (7.4)	942 (62)	37 (25–52)	1.9 (1.7–2.1)	101 (62–135)	128 (52–181)	390 (26)	37 (2.4)
Electrophysiological procedures	1321 (6.4)	694 (52)	26 (21–36)	1.8 (1.6–1.9)	–	–	860 (65)	47 (3.6)
PAPVC repair	808 (3.9)	356 (44)	38 (25–50)	1.8 (1.6–1.9)	50 (36–68)	81 (51–121)	36 (4.5)	3 (0.4)
Single ventricle-associated procedures	387 (1.9)	185 (48)	25 (20–31)	1.6 (1.5–1.8)	75 (47–134)	175 (123–244)	254 (31)	59 (15)
Transplants	108 (0.5)	63 (58)	23 (19–33)	1.7 (1.5–1.9)	131 (90–239)	205 (156–295)	61 (56)	21 (19)
Mechanical support implantation	97 (0.6)	32 (33)	25 (21–41)	1.7 (1.6–1.9)	93 (55–131)	127 (42–181)	43 (44)	66 (68)
Other procedures	511 (2.5)	271 (53)	30 (22–46)	1.7 (1.5–1.9)	25 (0–86)	90 (0–162)	159 (31)	43 (8.4)

* number of patients and percentage (%); ** median and interquartile range (IQR). BSA—body surface area; CC time—cross-clamp time; CPB time—cardiopulmonary bypass time; PAPVC—partial anomalous pulmonary venous connection.

**Table 2 jcm-09-02493-t002:** Hospital mortality according to procedural groups and main procedure leading to surgery.

Procedure	*n*. of pts	Hospital Mortality (%)
Total	20,602	622 (3)
**1-Septal defects repair**	5740	56 (1.0)
ASD	3735	30 (0.8)
VSD	1389	14 (1.0)
AVC (partial/intermediate)	593	8 (1.3)
ASD creation	23	4 (1.7)
**2-Right heart lesions repair**	**5542**	**166 (3.0)**
TV disease	2729	87 (3.2)
- TV plasty	2156	37 (1.7)
- TV replacement	410	39 (1.0)
- Ebstein’s repair	163	11 (6.7)
Conduit operations (RV/LV to PA operation, reop, other)	1173	35 (3.0)
PV disease:	775	13 (1.7)
- PV replacement	715	11 (1.5)
- PV plasty	60	2 (3.3)
RVOT procedure:	530	19 (3.6)
- PA reconstruction	252	10 (4.0)
- RVOT procedure	159	4 (2.5)
- DCRV repair	68	-
- 1 ½ ventricular repair	51	5 (1.0)
TOF repair	335	12 (3.6)
**3-Left heart lesions repair**	**4566**	**124 (2.7)**
AoV disease:	3408	68 (2.0)
- AoV replacement	1804	35 (1.9)
- Aortic root replacement	811	23 (2.8)
- Aortic stenosis sub-/supra-valvar	323	4 (1.2)
- AoV plasty	239	3 (1.3)
- Ross/Konno/Ross–Konno procedure	231	3 (1.3)
MV disease:	1158	56 (4.8)
- MV plasty	615	9 (1.5)
- MV replacement	538	47 (8.7)
- Supravalvular mitral ring	5	-
**4-Thoracic arteries and veins anomalies repair**	**1522**	**37 (2.4)**
Aortic aneurysm repair	609	10 (1.9)
Coarctation of aorta/aortic arch repair	384	7 (1.8)
CABG	276	10 (3.6)
Sinus of Valsalva aneurysm	63	2 (3.2)
Vascular ring repair	58	4 (6.9)
PDA closure	47	-
ALCAPA repair	38	1 (2.6)
Aortic dissection repair	26	3 (12)
Anomalous aortic origin of coronary artery repair	21	-
**5-Electrophysiological procedures (PM/ICD)**	**1321**	**47 (3.6)**
**6-PAPVC repair**	**808**	**3 (0.4)**
PAPVC repair	767	3 (0.4)
PAPVC scimitar repair	41	-
**7-Single ventricle-associated procedures**	**387**	**59 (15)**
**8-Transplants (heart/heart and lungs)**	**108**	**21 (19)**
**9-Mechanical support implantation (ECMO, LVAD, RVAD)**	**97**	**66 (68)**
**10-Other less common surgical procedures**	**511**	**43 (8.4)**

AoV—aortic valve; ALCAPA—Anomalous origin of left coronary artery from pulmonary artery; ASD—atrial septal defect; AVC—atrio-ventricular canal; CABG—coronary artery bypass graft; DCRV—double-chambered right ventricle; ECMO—extracorporeal membrane oxygenation; ICD—implantable cardioverter-defibrillator; LV—left ventricle; LVAD—left ventricular assist device; MV—mitral valve; PA—pulmonary artery; PAPVC—partial anomalous pulmonary venous connection; PDA—patent ductus arteriosus; PM—pacemaker; PV—pulmonary valve; RV—right ventricle; RVAD—right ventricular assist device; RVOT—right ventricle outflow tract; TV—tricuspid valve; VSD—ventricular septal defect.

**Table 3 jcm-09-02493-t003:** Associated procedures (*n* = 10,678 in 7564 patients).

Associated Procedure	*n* of Procedures
ASD repair	2844
MV plasty	709
Atrial arrhythmia surgical ablation	576
VSD closure	538
PV replacement	503
Aortic stenosis sub-/supra-valvar	489
TV plasty	479
Aortic root replacement	456
ICD/PM implantation–explantation	332
Pulmonary artery reconstruction	287
AoV plasty	274
AoV replacement	264
RVOT procedure	264
Conduit operation-reoperation	203
Aortic aneurysm repair	203
DCRV repair	198
PAPVC repair	178
Sinus of Valsalva repair	155
PDA closure	135
Aortic arch repair	120
ASD creation	89
PV plasty	85
Other annular enlargement procedure	55
Other coronary artery procedure	49
TV replacement	42
Cor triatriatum	37
Ventricular arrhythmia surgical ablation	36
ECMO implantation	33
Ebstein’s repair	28
MV replacement	16
Other less common procedures	1114

AoV—aortic valve; ASD—atrial septal defect; DCRV—double-chambered right ventricle; ECMO—extracorporeal membrane oxygenation; ICD—implantable cardioverter-defibrillator; MV—mitral valve; PAPVC—partial anomalous pulmonary venous connection; PDA—patent ductus arteriosus; PM—pacemaker; PV—pulmonary valve; RVOT—right ventricle outflow tract; TV—tricuspid valve; VSD—ventricular septal defect.

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
