# Peer review of "Surgery for Adult Patients with Congenital Heart Disease: Results from the European Database"

_jcm, 2020, doi:10.3390/jcm9082493_

Round 1

Reviewer 1 Report

This retrospective review of ACHD surgical cases is well organized and will add value to the ACHD literature.  I would suggest the authors review the work of Karamlou and colleagues (Ann Thorac Surg, 2010 Aug;90(2):573-9 and Circulation. 2008 Dec 2;118(23):2345-52) as it will inform their work.  I would be sure this is "the largest series of adult patients with CHD who required surgical" as I believe Karamlou study is of similar size and scope with very similar findings.

The manuscript does need a few improvements with grammar and consistency around abbreviations.

The only other comment is using the phrase "perfect candidates for VADs", (line 211).  I would remove the term perfect.

Overall I enjoyed reading this manuscript and think it is of value.

Author Response

ANSWER TO REVIEWER 1

This retrospective review of ACHD surgical cases is well organized and will add value to the ACHD literature.  I would suggest the authors review the work of Karamlou and colleagues (Ann Thorac Surg, 2010 Aug;90(2):573-9 and Circulation. 2008 Dec 2;118(23):2345-52) as it will inform their work.  I would be sure this is "the largest series of adult patients with CHD who required surgical" as I believe Karamlou study is of similar size and scope with very similar findings.

We would like to thank the reviewer for his comments on our manuscript.

Q1: The manuscript does need a few improvements with grammar and consistency around abbreviations.

A1: We have improved our manuscript and make some minimal changes regarding on abbreviations format.

C1: We provided to review and uniform the abbreviations as suggested.

Q2: The only other comment is using the phrase "perfect candidates for VADs", (line 211).  I would remove the term perfect.

A2: We reviewed the indicated sentence.

C2: At line 263 we removed the term “perfect”, as suggested.

Overall I enjoyed reading this manuscript and think it is of value.

Reviewer 2 Report

It is a retrospective study which evaluated the frequency, type and outcomes of cardiac surgery in adults with congenital heart disease (ACHD) utilizing data from European 26 Congenital Heart Surgeons Association Database. Several interesting observation were made such as a increase over time for right and left-heart lesions repair and a decrease for septal defects repair and transplant.

Below are my questions and comments.

  1. How is the age of patients change over the years?

2. Are there specific surgical procedures that tend to require re-operation?

3. Have similar studies been done and similar observations been made?

4. Please elaborate how the current data can serve as benchmark to validate current available risk score for ACHD requiring cardiac surgery.

Author Response

ANSWER TO REVIEWER 2

It is a retrospective study which evaluated the frequency, type and outcomes of cardiac surgery in adults with congenital heart disease (ACHD) utilizing data from European 26 Congenital Heart Surgeons Association Database. Several interesting observation were made such as a increase over time for right and left-heart lesions repair and a decrease for septal defects repair and transplant.

 We would like to thank the reviewer for his comments on our manuscript.

Below are my questions and comments.

Q1: How is the age of patients change over the years?

A1: A boxplot showing the variation of patients’ age at surgery during the last two decades has been elaborated. The median age remains fairly stable over the years, while the interquartile range and the maximum age widen, maybe due to the increase of life expectancy and quality of life over years.

C1: A figure, named as Figure 1, showing median age and intervals of patients’ age at surgery has been added in the manuscript (line 128-142). We added a sentence in the text (line 119-120).

Q2: Are there specific surgical procedures that tend to require re-operation?

A2: As shown in Table 1, we indicate the number of reoperations according to the main surgical procedural group

C2: No changes

Q3: Have similar studies been done and similar observations been made?

A3: According to current literature, we found a study made by Karamlou et colleagues on US data on ACHD population. They evaluated outcomes in terms of mortality according to type of hospital and type of cardiac surgery facility (as congenital and acquired). They found a better outcome for patients referred to a specialized congenital heart surgeons facilities with dedicated congenital heart surgeons. Our finding reinforce, by using a large series of European data the concept that a more consistent expertise with congenital heart problems and large volume centres are related to better outcomes (centre effect).

C3: We added this reference in the discussion section (line 292)

Q4: Please elaborate how the current data can serve as benchmark to validate current available risk score for ACHD requiring cardiac surgery.

A4:  With this study we provided consistent data by using a large European dataset on the incidence and mortality (as rough mortality) on adult patients requiring surgery for treating their CHD.

These data con be used to compare congenital cardiac surgical patterns of practice and outcomes. As a example other authors can compare the mortality by age groups, benchmark operations, volume load, CPB and CC times.

C4: no changes in the text

Reviewer 3 Report

Vida et al retrospectively investigated the frequency, type and outcomes of cardiac surgery for adult patients with CHD during the last two decades, using data from the European Congenital Heart Surgeons Association Database. The authors showed the trend of the surgical workload over the years for this specific subset of patients, indicating a progressive increase over time. They found a variation of the surgical scenario during the last two decades, characterized by an increase in right and left-heart lesions repairs and a significant decrease in septal defects repair and transplants. They concluded that operative mortality in adult patients with CHD remained low and stable over the last two decades, but some procedures continued to carry high mortality rates. This manuscript is well-written. The authors provided important data based on one of the largest series of adult patients with CHD who required surgical treatment. While of some interest, the following questions and comments are raised:

  1. (Methods) The reviewer was confused because there were “procedure”, “operation” and “patient” as the units in this study population. How did the authors count if the same patient had hospitalization and cardiac surgery multiple times during the two decades?
  2. (Methods) Patients with isolated acquired cardiac disease or receiving interventional cardiology procedures were excluded (n=1,869 patients). Was the number of patients counted accurately? The reviewer thought it was too small.
  3. (Methods and Results) The authors focused on surgical lesions. Confounding factors should be considered. Hospital mortality is strongly affected by the type of CHD, heart failure severity, or associated arrhythmias. Were these data not included in the European Congenital Heart Surgeons Association Database?
  4. (Results) There was no legend of Figure 4. The reviewer doubted if there is really an inverse correlation between center volume and percentage of death. Could the authors explain the statistical method and the meaning of the line in Figure 4?

Author Response

ANSWER TO REVIEWER 3

Vida et al retrospectively investigated the frequency, type and outcomes of cardiac surgery for adult patients with CHD during the last two decades, using data from the European Congenital Heart Surgeons Association Database. The authors showed the trend of the surgical workload over the years for this specific subset of patients, indicating a progressive increase over time. They found a variation of the surgical scenario during the last two decades, characterized by an increase in right and left-heart lesions repairs and a significant decrease in septal defects repair and transplants. They concluded that operative mortality in adult patients with CHD remained low and stable over the last two decades, but some procedures continued to carry high mortality rates. This manuscript is well-written. The authors provided important data based on one of the largest series of adult patients with CHD who required surgical treatment. While of some interest, the following questions and comments are raised:

 We would like to thank the reviewer for his comments on our manuscript.

Q1: (Methods) The reviewer was confused because there were “procedure”, “operation” and “patient” as the units in this study population. How did the authors count if the same patient had hospitalization and cardiac surgery multiple times during the two decades.

A1: In the ECHSA-DB each patient with a unique de-identified ID code. This allowed us to evaluate if the same patient had multiple cardiac procedures during the considered study period.

C1: We added a sentence on line 80-82.

Q2: (Methods) Patients with isolated acquired cardiac disease or receiving interventional cardiology procedures were excluded (n=1,869 patients). Was the number of patients counted accurately? The reviewer thought it was too small.

A2: The dataset we used was specifically addressed to surgical centers which deal with patients with congenital heart disease, so the number of patients excluded is certainly understimated, but that was the number we could calculate.  

C2: We add this as a limitation of the study at lines 308-310.

Q3: (Methods and Results) The authors focused on surgical lesions. Confounding factors should be considered. Hospital mortality is strongly affected by the type of CHD, heart failure severity, or associated arrhythmias. Were these data not included in the European Congenital Heart Surgeons Association Database?

A3: Data regarding preoperative and clinical variables were rather incomplete; we decided to use a minimal dataset in order to not affect our study’s statistical consistency.

C3: We added a sentence on lines 311-315.

Q4: (Results) There was no legend of Figure 4. The reviewer doubted if there is really an inverse correlation between center volume and percentage of death. Could the authors explain the statistical method and the meaning of the line in Figure 4?         

A4: In order to clarify the relationship between center volume and mortality we performed a logistic regression using the proportion of deaths as outcome variable and the number of operated patients as independent variable and an inverse relationship between center volume and mortality was observed. In particular, the results showed the intercept was negative and statistically significant. The corresponding hazard ratio was equal to 0.9997. The model fitted is presented in the figure, in which the number of patients is plotted in logistic scale to better appreciate the results. This generally means that the probability of death decreases with increasing of operated patients (i.e. volume center), but does not mean that all small centers have results not as good as the biggest hospitals. In the graphic the red line is the logistic regression model estimated.

C4: We have improved the legend of Figure 4, now named Figure 5, as suggested. An explanation of the statistical method has also been added in the text at line 186-192.

Round 2

Reviewer 3 Report

The authors promptly responded to the comments. The reviewer was satisfied with the answers to Q1-3, but felt that the answers to Q4 (legend of Figure 5 and statistical methods) were insufficient. The reviewer could not understand why the authors showed the relationship between center volume and mortality by using the intercept and hazard ratio. Not the intercept but the slope should be considered. The reviewer thought that spline curve might be appropriate.

Author Response

Q4: (Results) There was no legend of Figure 4. The reviewer doubted if there is really an inverse correlation between center volume and percentage of death. Could the authors explain the statistical method and the meaning of the line in Figure 4?

A4: We considered the slope and its exponential, the hazard ratio, to show the relationship between center volume and mortality. Spline regression is usually employed when data require a non parametric approach because of a not regular trend, which is not our case, in our opinion. Moreover, fixing the knots is necessary to estimate the spline, but looking at the figure, no breaking points are clearly visible. Finally, the order of the spline influences the estimated curve, in particular the starting and the ending points, which we are more interested in. For these reasons, we believe that logistic regression model is more appropriate to answer our hypothesis.

C4: We provided to correct the sentence, at line 187. No other changes.

Round 3

Reviewer 3 Report

The authors promptly responded to the comments. No more concerns.